# Interactive Search on the Web: The Story So Far

**Sareh Aghaei [1,\*], Kevin Angele [1], Elwin Huaman [1], Geni Bushati [1], Mathias Schiestl [1] and Anna Fensel [1,2,3]**

[1] Department of Computer Science, Semantic Technology Institute (STI), University of Innsbruck, 6020 Innsbruck, Austria; kevin.angele@sti2.at (K.A.); elwin.huaman@sti2.at (E.H.); geni.bushati@sti2.at (G.B.); csat8202@student.uibk.ac.at (M.S.); anna.fensel@sti2.at (A.F.)

[2] Wageningen Data Competence Center, Wageningen University and Research, 6708 PB Wageningen, The Netherlands

[3] Consumption and Healthy Lifestyles, Wageningen University and Research, 6706 KN Wageningen, The Netherlands

\* Correspondence: sareh.aghaei@sti2.at

**Abstract:** Search on the web, specifically fetching of the relevant content, has been paid attention to since the advent of the web and particularly in recent years due to the tremendous growth in the volume of data and web pages. This paper categorizes the search services from the early days of the web to the present into keyword search engines, semantic search engines, question answering systems, dialogue systems and chatbots. As the first generation of search engines, keyword search engines have adopted keyword-based techniques to find the web pages containing the query keywords and ranking search results. In contrast, semantic search engines try to find meaningful and accurate results on the meaning and relations of things. Question-answering systems aim to find precise answers to natural language questions rather than returning a ranked list of relevant sources. As a subset of question answering systems, dialogue systems target to interact with human users through a dialog expressed in natural language. As a subset of dialogue systems, chatbots try to simulate human-like conversations. The paper provides an overview of the typical aspects of the studied search services, including process models, data preparation and presentation, common methodologies and categories.

**Keywords:** search on the web; keyword search engine; semantic search engine; question answer system; dialogue system; chatbot; semantic web; natural language processing; review



## 1. Introduction

The immense amount of data on the web has created the need for search services to explore and fetch the desired information. With the widespread use of the internet of things and social networks, it is expected to have a further substantial increase in the amount of data generated [1,2]. Although estimation of the actual amount of data available on the web is difficult, there are some astounding numbers. In 2020, 1.7 MB (Megabytes) of data has been created every second by every person and 463 EB (Exabytes) of data will be generated each day by humans as of 2025 (https://www.weforum.org/agenda/2019/04/how-much-data-is-generated-each-day-cf4bddf29f/, accessed on 1 July 2022). So, with the huge amount of data being generated, it is irrefutable that finding the needed information on the web requires search services which embody extremely powerful and valuable tools for fetching any sort of information from the web [3].

A vast number of techniques and approaches have been proposed to provide search services on the web since the advent of the web [4,5]. Search services can be essentially viewed as answering machines. Search engines such as Google, Baidu, and Yahoo, with the most traffic [5], scour billions of pieces of the web and evaluate thousands of factors to determine which content is most likely to be relevant [6]. Here, keyword search engines, semantic search engines, question answering systems (QASs), dialogue systems, and chatbots can be considered as the main proposed types of search services.

To the best of our knowledge, there is no literature review dedicated to categorizing and discussing all search services on the web as well as the ways in which they are interrelated. Existing research focuses on either one type of search services (such as [7–9]) or two types of services (such as [3,10]). The various presented studies (Sections 3.1, 4.1, 5.1, 6.1 and 7.1) mostly conduct a detailed review of one aspect (such as mainstream techniques or categories) while our work investigates different directions (i.e., process models, data preparation and presentation, common methodologies and categories). Moreover, we offer a complete picture of the timeline of the evolution of search services which is missing in the literature.

Keyword search engines, as the first generation of search engines, are keyword-based and have been widely used under the hyperlink data environment [3]. They have adopted the keyword-based technique to find the web pages containing the query keywords and ranking search results [11]. Therefore, it is not easy to get an accurate result as they do not know the exact meaning of the keywords used [12].

Semantic search engines are meaning-based and have become known since the web 3.0 [13]. They make efforts to find meaningful and accurate results upon the meaning and relations of the words [14,15]. Classification of semantic search engines over user interaction mode include keyword-based semantic service, form-based semantic service, view-based semantic service, and natural language-based semantic service [16].

QASs can give users precise answers to the questions presented in natural language [17]. Given a user's natural language question, the system converts the question to a query and then submit the query to the search engine. Afterward, the system extracts all relevant answers from the search results and finally selects the most desired answers to return [18]. QASs fall into different groups based on the type of answer, including sentence/paragraph-answer based QAS, yes/no-answer based QAS, multimedia-answer based QAS, opinionated-answer based QAS, dialogue-answer based QAS [14].

Dialogue systems (also referred to as conversational systems) are a subset of QASs targeting to interact with humans through a dialog expressed in natural language. The process model of a dialogue system is composed of three layers: user experience, conversation engine, and data layer. The conversation engine layer, as the middle layer of the model, include natural language understanding, dialogue manager, and natural language generation [19]. The dialogue systems can be divided into two groups, namely, task-oriented and non-task-oriented systems, from the aspect of having a task as the goal or not in the dialogue [20]. So the task-oriented dialogue systems target assisting users in completing tasks, while non-task-oriented dialogue systems such as chatbots aim to make conversation with human beings.

Chatbots are designed to simulate human-like conversations [9]. Chatbots' early and primary role has been constructing extended conversations to mimic chats characteristic of human interaction. However, chatbots have been employed to do practical tasks over time [10,21]. Chatbots can be divided into three groups upon goals being achieved: informative chatbots, chat-based chatbots, and task-based chatbots [10].

The interaction of the described search services including keyword search engines, semantic search engines, QASs, dialogue systems and chatbots is depicted in Figure 1 (e.g., QASs can overlap in some aspects with semantic search engines or dialogue systems are a subclass of QASs).

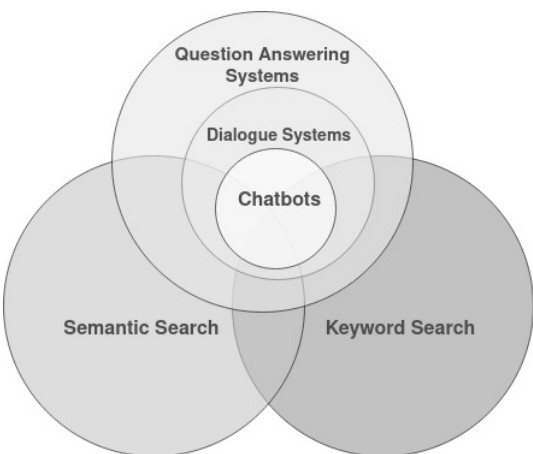

**Figure 1.** The interaction of different types of the search services.

This paper presents the evolution and typical aspects of each search service such as process models, data preparation and presentation, common methodologies, categories (depicted in Figure 2) and clarifies their interactions. For each search service, numerous research works have been introduced and developed since the advent of the web which can not be described exhaustively in one paper. Therefore, the most typical development trends are reviewed over time in this paper.

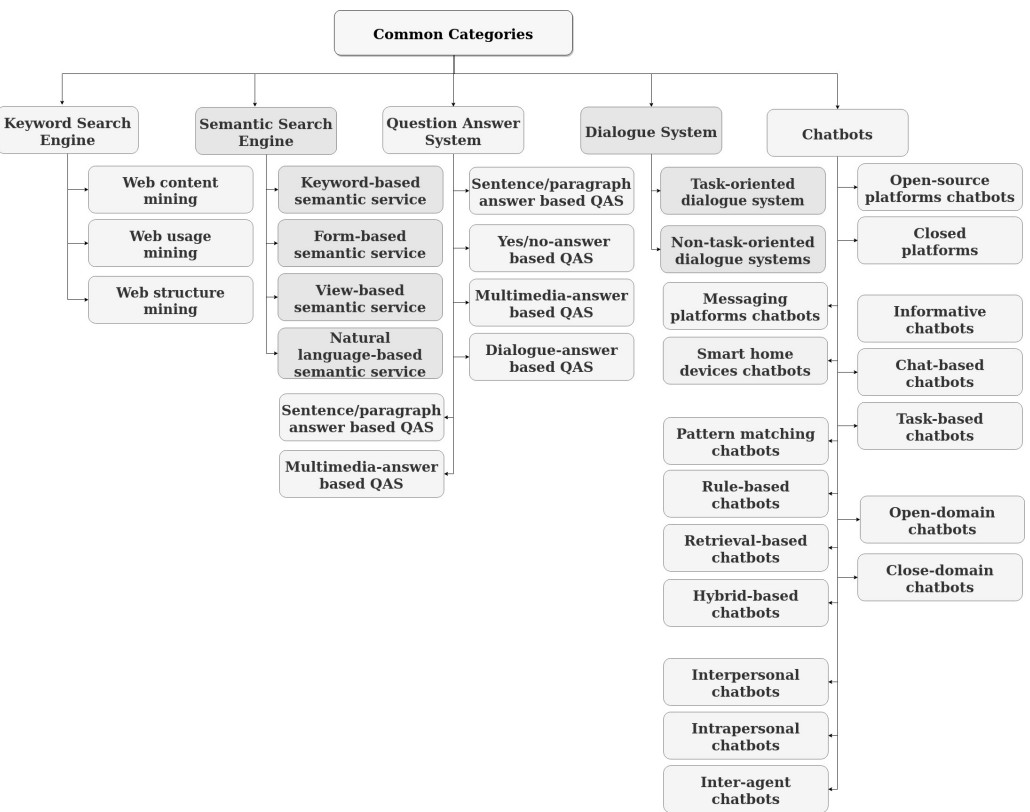

**Figure 2.** The common categories of different types of the search services.

The remainder of the paper is organized as follows. The survey methodology is given in the next section. In Section 3, keyword search engines are described. Then, semantic search engines are discussed in Section 4. Section 5 describes QASs. Dialogue systems and chatbots are explained in Sections 6 and 7, respectively. The future of search on the web is discussed in Section 8. Section 9 describes the major findings of the research study. Finally, Section 10 concludes the paper.

## 2. Survey Methodology

For this paper, a typical methodology for doing a survey has been followed. The peer-reviewed publications as the primary resource have been identified using the scholarly indexing services: Google Scholar, IEEE (Institute of Electrical and Electronics Engineers) Xplore, ACM (Association of Computing Machinery) Digital Library, Scopus, and DBLP (Digital Bibliography & Library Project). The query terms included keywords "search engines", "semantic web", "semantic search engines", "knowledge graphs", "question-answering systems", "dialogue systems", "chatbots". The selected time frame to find relevant resources was mainly from 2000 to the present, as only keyword search engines were extensively used before. Authors and affiliations of the publications were also used as keywords to find additional relevant sources of knowledge. Moreover, the citation count of the papers has been a key criterion to select and review those. Based on the citation count, for each subsection, the most known papers were examined and reviewed.

In order to provide a comprehensive investigation for each type of search service on the web, the search services, including keyword search engines, semantic search engines, QASs, dialogue systems, and chatbots, are reviewed in separate sections. Each section consists of six subsections: current state of the art and related works, process model, data preparation and representation, common methodologies, categories, and summary.

Firstly, a short history and definition of search services is given and its interaction with the other types of search services is shown. Then the current state of the art and the related works are described in the following subsection. The process model subsection presents a high level conceptual model which defines the structure and behavior of the search service. Next, data preparation and representation are described. The primary methodologies to develop the search service are expressed in the common methodologies subsection. In the categories subsection, the major groups of the search service are indicated, and finally, a summary of the search service is given in the summary subsection.

## 3. Keyword Search Engine

In the early years of the web, documents were indexed manually by making a list of links hosted on specific web servers. For example, Archie, as the first internet search engine in 1990, was an index of FTP (File Transfer Protocol) files that allowed users to create simple requests for searching files [22]. Due to the increasing growth of the documents on the web, the indexing approach quickly was replaced by keyword search engines.

Keyword searching is the most common form of text search on the web by creating text queries and retrieving information using these keywords. The keyword search engines have been widely used on the web 2.0 to explore the documents by considering the links to/from a web page as a method of determining relevance [13]. Keyword search engines have been known as syntactic search engines because they depend on keywords as text in their queries [3]. Some examples of syntactic search engines include Google, Yahoo, Ask, and Msn [22].

Inline with keyword search engines, metasearch engines target to pass queries simultaneously to multiple search engines and then collect and integrate their results to a single result listing at the same time avoiding redundancy [23,24]. For example, Metacrawler [25] combines the results from some of the search engines such as Google, Yahoo, Ask, Msn instead of crawling the web and maintaining its own index of documents. In contrast to metasearching which is based on just-in-time processing, federated searching is based on just-in-case processing [26]. Federated searching combines a large amount of data into a single repository that can be searched [27]. For example, Google Scholar as a federated searching system provides a single user interface for searching across scholarly information [26].

Consider the case of finding and ordering food on the web through the keyword search engines. The keywords such as "order food online", "restaurants in Innsbruck", "free delivery" have been required to be searched by users looking for free deliverable meals in Innsbruck and then only the web pages including the keywords have been returned.

Next, the users had to check the returned pages one by one to find their desired restaurants and finally order the foods.

### 3.1. Current State of the Art and Related Works

Keyword search engines paved the way for the next generations of search engines. It is difficult to get an accurate result in keyword search engines as they do not know the exact meaning of the keywords used. So, polysemy words and synonym words lead to false positive and false negative, respectively. With the massive amount of data available on the web, a simple keyword text search is an ineffective solution, and intelligence should be embedded into search engines. Recent efforts are being made to provide an effective way to search the web that will be treated in Section 4 as semantic search engines.

The history and rise of keyword search engines are studied by Seymour et al. [22]. Here, various index-based tools and keyword search engines, including Archie, Gopher, Veronica, Judgehead, Aliweb, Altavista, Ask Jeeves, and Northern Light, continue to Google, Yahoo! and Bing are described [28].

In Rahman's survey [29], the major challenges, issues, and downsides of keyword search engines have been discussed in detail. Then, the paper has introduced different non-keyword based approaches, including semantic search, concept-based search, exploratory search, content-based search, open domain QA and a computation knowledge engine to address the keyword searching obstacles. The survey presented by Selvan et al. [30] offers a review of the ranking algorithms that the keyword search engines use. According to the paper, the ranking algorithms are categorized into three categories, including link analysis, personalized web search ranking, and page segmentation algorithms. Further, the paper gives an overview of the most prominent ranking algorithms of each category. The Page Rank (PR) [31], Hyperlink-Induced Topic Search (HITS) [32], and Focused Rank [33] as the main link analysis algorithms, the integrated page ranking algorithm as the key personalized web search ranking algorithm, fixed-length page segmentation, document object model-based page segmentation, vision-based page segmentation and combined approach segmentation as the major page segmentation algorithms have been discussed in the survey.

According to studies by Tokgoz et al. [34], Hussain et al. [35] and CheshmehSohrabi et al. [36], Yahoo outperforms Google in image retrieval (users use one or more keywords to retrieve the relevant image or images they are looking for) while studies by Uluc et al. [37], Cakir et al. [38] and Adrakatti et al. [39] reveal contradictory results on the effectiveness of image search engines and indicate the outperformance of Google in image retrieval as compared with Yahoo.

### 3.2. Process Model

Although different keyword search engines have various process models, they are essentially composed of three main components including information collection component, indexing component and ranking component as shown in Figure 3:

1.  Information Collection Component: web crawlers are responsible for collecting information. A crawler (also called spider or bot) can be assumed as a program that continually browses the web to collect new pages. The collected pages are used in the indexing in the next component.
2.  Indexing Component: the collected web pages are stored, organized, and indexed in this component to improve the speed of retrieval. When a page is in the index, it is in the running to be displayed as a result of relevant queries.
3.  Ranking Component: this component provides the pieces of content that evaluate the desired pages for the query, which means that results are ordered by most relevant to least relevant. The ranking algorithms which can be applied in the ranking component are described in Section 3.4.

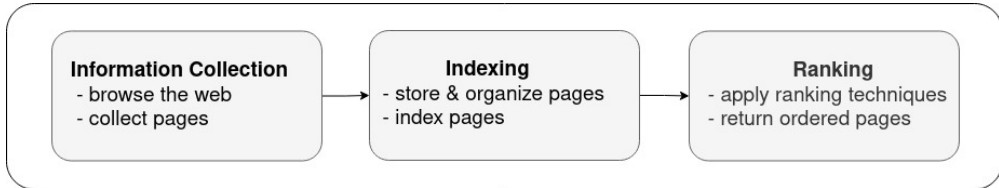

**Figure 3.** Keyword search engine process model.

*3.3. Data Preparation and Representation*

Keyword search engines have been broadly applied in the first and second generations of the web. The web 1.0 can be considered as the first generation of the web, which was ready-only, static, and somewhat mono-directional. The web 1.0 included websites with static HTML (HyperText Markup Language) pages that updated infrequently. The web 2.0, as the second generation of the web is a read-write web with blogs, RSS (Really Simple Syndication), wikis, mashups, tags, folksonomy, and tag clouds as its major technologies and services [13]. Therefore, the documents on the web, either HTML or XML (Extensible Markup Language), can be considered the primary underlying resources for the keyword search engines, which are generally based on the occurrence of words in the documents and do not determine the intent and contextual meaning of the words.

*3.4. Common Methodologies*

With the increasing number of documents, search engines have the aim to find the best ranking order to satisfy user's query [40]. This is a task done by ranking algorithms. Ranking models assign scores to the documents representing relevance and similarity between the documents and the user queries. In general, the existing ranking techniques can be organized into three groups, as follows:

1. Content-based ranking: these approaches rank the relevant pages based on pages' content and keywords. Firstly, the words from the user query are stripped down to the root. The root words and their synonyms are considered for the construction of a dictionary. Then the keywords of each page are compared against the dictionary. Accordingly to the matches found, the relevancy of each particular page against the user query is computed as the ranking score.

2. Usage-based ranking: these approaches aim to rank web pages based on users' past navigation and retrieval patterns. A ranking score is assigned to each page which indicates how often they are viewed on the web. Thus, it determines the page's relevancy by its selection frequency. Ranking based on usage only can not guarantee precise results due to the other indications such as time spent on reading the page, the number of times the page was saved/printed or added to the bookmarks, and the actions of following the links of the page are neglected.

3. Link-based ranking: these approaches compute the ranking scores based on the links between the pages. For example, from the fact that a page has many links and references, it is derived that it must have something interesting to express. The link-analysis-based algorithms are generally computed offline, even before receiving any query from the user. Here, the popularity of pages is calculated by building a graph using a set of nodes and analyzing the existing links in it. The PR algorithm and the HITS algorithm are the most common examples of linked-based ranking algorithms.

*3.5. Categories*

In general, keyword search engines are divided into the following categories:

1. Web content mining: these search engines mine the content of web pages to extract the result by performing different mining techniques and shrink the search data, which become easy to find required user information [41].

2.  Web usage mining: according to the log information stored during user interactions while surfing the web, user navigation patterns are discovered. Then the discovered patterns are applied to rank and fetch the desired pages against user queries [42].
3.  Web structure mining: the main idea behind web structure mining search engines is to discover the structure of web pages based on the hyper links among them, create a structural summary and finally use the created the structural summary to extract the pages of given keywords [41,42].

*3.6. Summary*

Keyword searching is the most common form of text search on the web. The keyword search engines, namely syntactic search engines, create the text queries and retrieve information using the keywords. There are three key steps to how most search engines work, including crawling, indexing, and ranking. To return the web pages in an ordered manner, web page ranking methods are applied, which can rank the pages in order of their relevance based on ranking algorithms. The different ranking techniques used in the keyword searching engines are divided into three approaches: content-based ranking, usage-based ranking, and link-based ranking. The PR and HITS as the link-based ranking algorithms are widely used in search engines. Web content mining, usage mining, and structure mining are assumed to be the major groups of keyword search engines based on their applied ranking techniques. Since the keyword search engines can not intelligently understand the context of what is being searched, they cannot be seen as an intelligent and effective solution to retrieve the information on the web.

**4. Semantic Search Engine**

Currently, most of the web's content is published so that humans can read it, and machines can identify where the main parts of the content are a text, a picture, or a link. However, machines cannot understand the semantic meaning behind them. Moreover, the increasing volume of data and web pages on the web has shown that traditional search engines are less suitable to provide the correct answer for users' questions. For example, a search engine retrieves information from the web, based on syntax matching of keywords provided by a user. This means that a keyword-based search engine will retrieve hundreds of results, which are mostly not relevant to the query.

Furthermore, some limitations of traditional search engines are the lack of a structure when representing information (information is presented in a non-machine-readable format), heavy usage of computational power, and insufficient quality when answering long and complex queries. To overcome these problems, semantic search is required to improve search accuracy by consuming machine-readable web pages and semantic annotations.

With the growth of semantic web, semantic search engines have been fulfilled to search on the web [43]. The primary idea of semantic search stems from a data searching technique in which a search query intends to fetch keywords and specify the intent and contextual meaning of the user's query. Semantic search engines return relevant data to answer complex queries, which are restricted to particular entity types. For example, a query "Innsbruck restaurants serving Italian pizza" must be split into different entities such as "Innsbruck", "Restaurant", "Pizza" that represent a city, place and food, respectively. In other words, semantic search enables searching for entities (e.g., things, persons, places) instead of just strings as keywords. So, in the case of table order, semantic search improves traditional search by relying on data that come from the semantic web and are supported by semantic web technologies [43,44].

*4.1. Current State of the Art and Related Works*

In this section, we discuss work that has been done on semantic search engines. Ilyas et al. [45] propose an abstract conceptual process model for semantic search engines, which should consist of ontology manager, web crawler, query builder and preprocessor, and inference engine. In the work presented by Sánchez-Cervantes et al. [46], a meta-

search system, called LINDASearch, is introduced to provide information about the well-known open linked data projects (e.g., DBpedia). LINDASearch aims to overcome several limitations, including faceted navigation and data unification over various domains, time optimization, and results' scalability. Sahu et al. [47] compare several search engines, for instance, evaluating search queries and their time on retrieving the answers, as well as the precision of the answers. The authors state that Google has the best performance, followed by Yahoo and Bing, respectively. Hussan [48] and Jain et al. [49] survey semantic-based search engines and point out their pros and cons. The authors state that all the surveyed semantic search engines use mainly semantic web technologies. They are more accurate than keyword-based search engines. However, semantic search engines are still complex to be implemented. For instance, semantic search engines must focus on hybrid approaches that combine various ways of interactions, leveraging their advantages [16].

Furthermore, Uren et al. [16] focus on reviewing the user experience of semantic searches. They describe four ways a user can interact with semantic search systems: keyword-based, form-based, view-based, and natural language-based. Some limitations are mentioned for form-based and view-based approaches, such as those that do not allow relation-based search and have low performance on showing large ontologies on an interface. In addition, the keyword-based search is limited to mainly using syntactic matching techniques, and natural language-based search provides means to formulate long-tail queries yet require more user interactivity.

Additionally, some limitations for semantic search engines are: the knowledge acquisition process limits the powerfulness of semantic search engines [50], and furthermore, there is some work to be done on matching and ranking methods used on semantic search engines, which mostly reuse information retrieval techniques [16]. Last but not least, search engines must rely on high-quality knowledge sources [51]. However, it is not clear to which degree knowledge sources are correct or complete, e.g., Google's search engine relies on Wikidata, which might contain errors, duplicates, or missing values.

*4.2. Process Model*

There are several semantic search engines developed from the need for efficient, accurate, and scalable search on the web, e.g., some instances of semantic search engines include Kosmix, Kngine, Hakia, Cognition, Falcons, Lexxe, Scarlet, Sindice, Swoogle, SWSE (Semantic Web Search Engine), and Watson [49,52,53], while all these semantic search engines are developed following specific use cases and are based on different assumptions, a mutual understanding of the model that relates them to each other exists. For instance, (a) semantic web documents discovery, (b) indexing, (c) analysis, and (d) interface are the components of Swoogle [54], (a) Resource Description Framework (RDF) crawler, (b) document analysis, (c) vocabulary identification, reasoning, and indexing, (d) summarization, and (d) interface are modules of Falcon [55], and (a) Crawler, (b) Query detection and extraction, (c) ontology analysis, (d) query indexing, (e) query processor, (f) ranking, and (g) interface are components of Hakia [49].

The semantic web allows structuring content on the web to enable machines to perform tasks that need a level of data interpretation. Semantic search engines are mainly based on semantic web technologies, including standard data modeling, syntax, schema, and query languages and protocols. These technologies make semantic search engines work. In the following, semantic search engines process model (see Figure 4) is described:

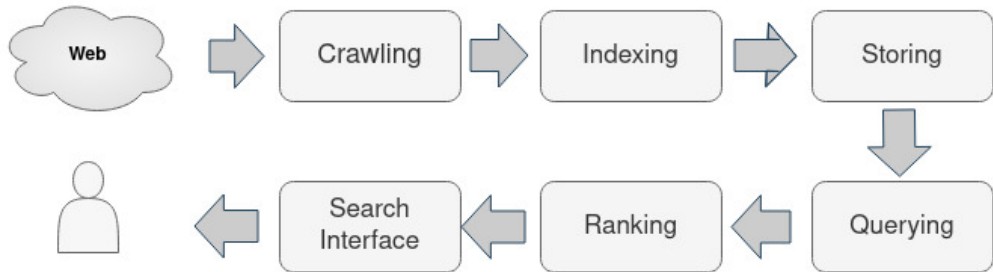

**Figure 4.** Semantic search engines process model.

1. Crawling: a crawler aims to discover documents and collect them. Like classic search engines, semantic search engines collect data using crawlers, for instance, crawling RDF documents embedded on websites. For that, more specialized crawlers have been developed [56], which provide various methods for crawling data; (1) direct URL (Uniform Resource Locator) crawling that bootstraps the crawling process, (2) Google-based crawling, which retrieves hyperlinks directly from Google search engine, (3) extracting and following hyperlinks limited to a certain depth and threshold, (4) RDF crawling that fetches semantic annotations embedded on webpages [45], furthermore, (5) crawlers that combine previous approaches, for example, using Google-based crawling to retrieve hyperlinks for most common words used in a specific language (e.g., English language).

2. Indexing: the indexing process analyses the retrieved documents from the crawling process. For instance, the vocabulary used and the relationship between resources and metadata about the retrieved documents (e.g., last modified) are analyzed. For instance, Swoogle uses a rational surfer model to rank retrieved documents [49].

3. Storing: the indexed documents are later on stored in a knowledge base in the form of graphs, which contain triples (subject, predicate, object). The goal of storing the data using semantic technologies is to facilitate the answer of complex queries [49]. For instance, there are several graph database systems, such as GraphDB (https://www.ontotext.com/products/graphdb, accessed on 1 July 2022) and Neo4J (https://neo4j.com/, accessed on 1 July 2022), which support semantic queries.

4. Querying: after the indexing and storing processes have been done, the knowledge can be queried, while querying to traditional search engines return documents, the result of a semantic web search query is richer than just simple documents. Semantic search engines return a representation of an entity (note that an entity or a resource includes web pages, parts of a web page, devices, people and more) (i.e., classes, properties, and literal values). For this reason, the query system provided by a semantic search engine should be able to perform complex search queries, e.g., a user can express the context for a term that he or she is looking for, and the semantic search engine can disambiguate the term [45].

5. Ranking: the ranking process runs semantic analysis and concept match between the user's query and the output produced by the semantic search engine [49]. The ranking process evaluates several semantic and statistical metrics, e.g., context and popularity, which improve semantic search engines' algorithm [57]. For instance, Anyanwu et al. [57] propose SemRank that applies several techniques (e.g., semantic association) for analyzing and ranking relationships between two instances in a knowledge base.

6. Search Interface: "the proof of the pudding is in the eating", providing means to access the knowledge base is a crucial part of a semantic search engine. For instance, it is essential to provide means that facilitate access to knowledge for humans and machines. Services like REST (Representational State Transfer) service API (Application Programming Interface) [54], keyword-based, form-based, view-based, or natural language-based semantic search systems must be supported [16].

*4.3. Data Preparation and Representation*

A semantic search engine collects, indexes, and stores richly structured data sources to later provide querying mechanisms to explore and retrieve knowledge. Semantics enables machines (e.g., semantic search engines) to understand the represented data better and discern the entities and their relationships.

Currently, most of the web's content is in natural language text, which raises an open question of whether computers will ever become as fluent as humans in understanding natural language text to interpret it the way humans would, for example, understanding entities and relationships between them. In order to overcome this problem, semantic technology techniques for retrieving and transforming structured and unstructured data into a knowledge base paradigm are required. For instance, various standards, such as RDF (https://www.w3.org/RDF/, accessed on 1 July 2022), RDF Schema (RDFS) (https://www.w3.org/2001/sw/wiki/RDFS, accessed on 1 July 2022), and Web Ontology Language (OWL) (https://www.w3.org/OWL/, accessed on 1 July 2022), were developed for the syntax and data model [58].

RDF has been published as a standard model for data interchange and proposed as a graph-based data model. In RDF, a document states that particular entities have properties with specific values. These are known as triples (subject, predicate, object). For example, the "Innsbruck Restaurant's phone number is +41 672343247" statement can be expressed as follows: a subject denoting "Innsbruck Restaurant," phone number as the predicate or property, and the object as value "+41 672343247".

Schema.org (https://schema.org/, accessed on 1 July 2022) is the most widespread vocabulary and the de facto standard for annotation of data on the web. It is supported by the major search engines Bing, Google, Yahoo, and Yandex since 2011. Furthermore, Schema.org vocabulary, along with the Microdata, RDF in Attributes (RDFa), or JavaScript Object Notation for Linked Data (JSON-LD) formats, are used to markup not only data but also content and services on the web.

Semantic annotations have gained attention since the introduction of Schema.org, which empowers web search on a global scale. Search engines can recognize semantic annotations because they mark up websites' content and are embedded on websites. Fensel et al. [59] state that semantic annotations are the basis for building a knowledge graph.

Knowledge graphs, also known as knowledge bases, are large semantic nets that integrate diverse sources to represent knowledge in target domains [59]. Google's knowledge graph, launched in 2012 initially to improve Google's search results, boosted the adoption of knowledge graphs. For instance, large technology companies, including Amazon, Facebook, Google, Microsoft, and many more, have knowledge graphs, and have invested in their curation with the purpose to improve their web-scale services (e.g., knowledge graphs' content can be easily explored and analyzed via semantic search engines) [60].

Technologies that empower semantic search engines are already in place. The next step is to promote their adoption by everybody who produces content on the web. The performance of semantic search engines will increase as more machine-readable data become available on the web.

*4.4. Common Methodologies*

There are several semantic search engines developed from the need for efficient search engines, while all the semantic search systems differ, some common methodologies relate them to each other.

1.　Knowledge acquisition: To effectively harvest structured and unstructured knowledge from the web, hybrid crawlers were used. For instance, crawling semantic annotations from websites, following links from the crawled semantic annotations, harvesting RDF/XML documents, and URLs using traditional search engines. Furthermore, several methods for transforming unstructured and semistructured data into structured knowledge are needed, e.g., converting comma-separated values (CSV) data into RDF.

2. Knowledge base construction: It summarizes methods, such as schema alignment, entity matching, and entity fusion, for integrating knowledge into a knowledge base. For instance, to detect duplicates (i.e., entity matching), it is necessary to compare every entity with each other, which is not recommendable for large knowledge bases [61,62]. In this case, indexing techniques might help to reduce the number of comparison, e.g., some indexing approaches are: an ontology-based index [63] that stores the ontology graph, an entity-based index that takes into account the relationships between entities, and a textual-based index that considers triples (subject, predicate, object). Additionally, more index techniques are listed in the paper presented by Lashkari et al. [64].

3. Semantic search services: Approaches to capture and process search queries based on various techniques, such as entity ranking algorithms, e.g., ranking approaches can be classified into three categories [65]: entity, relationship, and document ranking. Furthermore, semantic search services comprises the coupling between the stored knowledge base and an ontology to support the generation of queries, e.g., traversing the knowledge base (or a part of it) and using query templates are common approaches used for refining/creation of queries.

4. Semantic search presentation: Common ways of interacting with semantic search engines involve a) services for machines like REST service APIs and b) user-friendly interfaces such as keyword-based, form-based, view-based, and natural language-based semantic search systems.

*4.5. Categories*

According to Uren et al. [16], semantic search engines can be categorized into four groups based on their user interaction mode:

1. Keyword-based semantic service: These search engines try to boost the performance of conventional keyword search engines by considering semantic entities that match query keywords. They translate query terms into semantic entities through typed links among entities on the data web to find more accurate and relevant information.

2. Form-based semantic service: Form-based semantic search services aim to guide users to make queries based on the information needed. They facilitate formulating semantic queries by translating ontologies' parts into forms, menus, and drop- down lists.

3. View-based semantic service: The services intend to help users construct queries and explore domains by ontology presentation and navigation. The considerable benefit of web-based semantic services is that users can easily understand the domain. The query vocabulary and content classification scheme can be presented in intuitive formats.

4. Natural language-based semantic service: the main idea is to employ semantic markup and natural language processing techniques in question-answer systems. The service takes a query expressed in natural language and a given ontology as input and finds the answer from one or more knowledge bases that subscribe to the ontology. Therefore, they are more flexible than form-based and view-based semantic services and do not require users to learn the vocabulary or structure of the ontology to be queried.

Furthermore, Wei et al. [66] classify semantic search into six categories: (i) document-oriented search that is an extension of conventional information retrieval techniques and retrieves semantic annotations, (ii) entity and knowledge-oriented search, which improves the previous category by exploiting links between entities to retrieve additional knowledge, (iii) multimedia information search that allows retrieving semantically related multimedia (e.g., images), (iv) relation-centered search, which additionally pre-processes user's query to find out relations between query terms, furthermore, relation-centered search are most often exploit in QASs (see Section 5) semantic analytics that entirely takes advantage of knowledge bases to discover and interpret complex relations between entities, and (v) mining-based search, which infers new assertion based on a knowledge base.

*4.6. Summary*

Semantic search engines are the cherries on the cake! They show the benefits of using semantic web technologies on specific tasks. We discussed a general overview of the common components/processes of semantic search engines: crawling, indexing, storing, querying, ranking, and search interface. It should be noted that current semantic search engines have differences in the way they implement the components. Additionally, we point out that the technology (e.g., RDF, Schema.org) that empowers semantic search engines is already in place, and we only need to promote their adoption. Furthermore, a classification of semantic search engines over user interaction mode, and task-oriented search has been discussed. Last but not least, we list some limitations or challenges that must be faced, such as heterogeneity of information (i.e., ontology quality), scalability (i.e., the rapid growth of content on the web), and quality (i.e., a trade-off between correctness and completeness of data).

## 5. Question Answering System

QASs can be viewed as an extension of search engines in the sense that they target finding precise answers to natural language questions, rather than returning a ranked list of relevant sources [67]. Since in natural language, the same meaning can be expressed in different ways and the same phrase having different meanings, QASs need to address lexical gap and ambiguity, respectively [68,69]. For example, the vocabulary used in a question can be different from the one used in the underlying knowledge source. The first QASs were introduced in the late 1960s and early 1970s to access data over databases [67,70,71]. With academic research, QASs have become a crucial topic and attracted massive attention over unstructured and structured data such as text documents and RDF knowledge graphs. Thanks to the development of the semantic web, a large amount of structured data has become available on the web, and the demand for QASs increases day by day [72].

Imagine the example of table order on the web through QASs. These search services allow users to express their questions in a natural language form such as "Which restaurants in Innsbruck serve Austrian cuisine on Sundays for between €30 and €60?" and then the precise answers are returned to the users. Here, users do not get lost in massive resources on the web and only receive answers of their posed questions over knowledge resources.

QASs overlap in some aspects with semantic search engines and even can be considered as a kind of natural language semantic search engines which employ semantic mark-up and natural language processing techniques to provide effective and convenient query techniques for end users [16].

*5.1. Current State of the Art and Related Works*

With the extensive research of the past years on QASs, these systems are used in a wide range of application areas, including web communities, medicine, industry, culture, or tourism [70,71]. The main focus of past surveys can be summarized as follows.

The survey presented by Dimitrakis et al. [71] obtains insights regarding the underlying knowledge source in QASs, where their knowledge source can be (1) structured data such as SQL (Structured Query Language) databases and knowledge graphs, (2) unstructured data in the form of text documents, or (3) mixtures of them. It also provides several aspects for structuring the landscape of QASs and labeled the main recent systems based on different aspects (knowledge source, types of questions, and domain type). Furthermore, the most common evaluation datasets have been grouped according to various criteria, including the domain's type, knowledge source, available tasks, and evaluation metrics.

Analyzing QA collections to provide better descriptions about the challenges of QASs has been the main focus in the work presented by Rodrigo et al. [73]. It has shown that QASs can hardly address various rewordings in questions and documents and infer information that is not explicitly mentioned in texts. Further, a set of directions for future evaluations has been suggested.

The study conducted by Wu et al. [74] gives a comprehensive review of knowledge base QA approaches. The methods to answer the question over a knowledge base have been classified into semantic parsing and information retrieval. Semantic parsing is a high-level analysis that targets to convert unstructured natural language questions into logical forms or executable queries. Information retrieval aims to extract information from questions, detect candidates across the knowledge base, and finally find the most suitable answer among the candidates. The study has discussed the mainstream techniques of each category, similarities, and differences among them in detail.

*5.2. Process Model*

According to the type of the underlying knowledge sources, the process models of QASs can be classified into two broad categories, namely: document-oriented model and data-oriented model [14,17].

The document-oriented model targets extracting the answer from plain text and employs traditional information retrieval techniques combined with machine comprehension methods [71]. QASs over documents adhere to a pipeline model with three main modules, including question analysis, passage retrieval, and answer extraction as shown in Figure 5 [14,71], and can be described as follows:

1.  Question analysis: as the first activity, questions are classified based on their type, leading to recognizing the expected answer types. Then named entities of input questions are identified, and their relations are detected. The last activity of this module aims at enhancing question phrasing through adding more descriptive information to increase the accuracy of the system, for example, using WordNet as a lexicon in order to retrieve semantically equivalent information [14,71].

2.  Passage retrieval: to find relevant information in the underlying documents as source knowledge, this module employs information retrieval techniques for returning a ranked subset of the most relevant documents. Then the relevant documents are segmented into shorter units, namely, passages. Finally, the candidate passages are ranked according to features such as the number of question words in the passage and the number of named entities having the answer type in the passage. These features should determine the probability of containing the precise answer [14,71].

3.  Answer extraction: this module targets identifying the answer candidates from the ranked list of passages. Then the candidate answers are ranked according to features that reflect the probability of being the precise answer. To deal with the partial answers between passages or documents, it is responsible for generating the final answers and computing a confidence score that reflects the confidence of its accuracy.

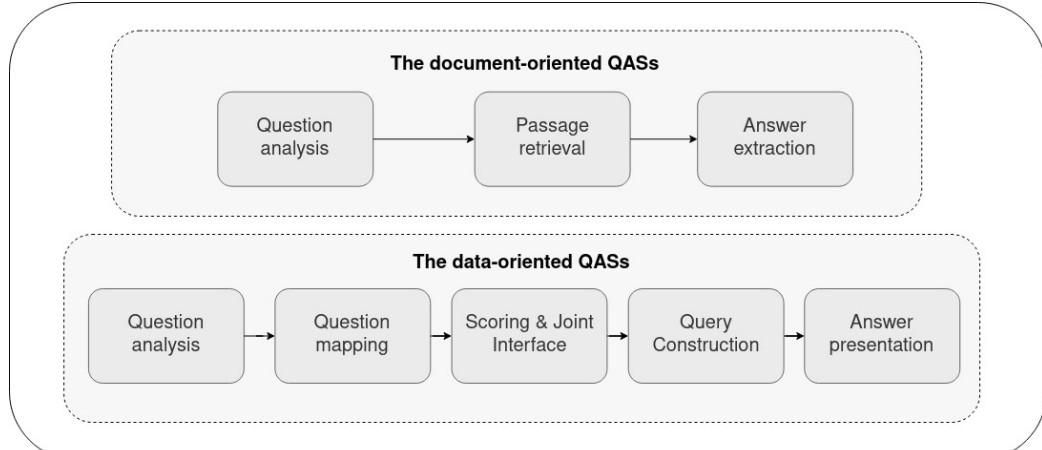

**Figure 5.** The document-oriented process model.

With the increasing maturity of structured data on the web, e.g., knowledge graphs, most QASs over structured data adopt a multi-component process model.

The major components of the data-oriented process model can be listed as question analysis, question mapping, scoring, and joint interface, query construction, and answer representation as shown in Figure 5 [67].

1.  Question analysis: it aims at analyzing the input question linguistically and syntactically. The linguistic analysis of the question leverages part-of-speech taggers and parsers to capture the syntactic structure of the question, e.g., Named Entity Recognition. The semantic analysis targets identifying the question type and the focus question.
2.  Question mapping: this component's main goal is to match question words or phrases to their counterparts in the underlying knowledge source, e.g., RDF knowledge graphs. Due to the lexical gap and the ambiguity between user questions and the underlying knowledge graph vocabulary, synonymy, hypernym and hyponym should be considered in computing similarities.
3.  Scoring and joint interface: to select only one candidate among the candidate components, a scoring mechanism is required to score the candidates. The semantic similarity between the candidates and the question can be applied to achieve a scoring mechanism to define a matching score. Moreover, string matching combined with linear programming can be used.
4.  Query construction: in order to transfer the question to an executable query such as SPARQL Protocol and RDF Query Language (SPARQL) (https://www.w3.org/TR/rdf-sparql-query/, accessed on 1 July 2022), the approaches can be summarized into two groups: (a) template-based approaches which map input questions to generated SPARQL query templates, (b) template-free approaches which aim at creating SPARQL queries according to the given syntactic structure of the input question.
5.  Answer presentation: since the structured representation of answers is not intelligible for users, the answer presentation component follows some processing activities to transfer the RDF answers to a natural language form.

*5.3. Data Preparation and Representation*

The fundamental goal of QASs is supplying accurate answers to questions posed by users in a natural language form [71,75]. The knowledge sources which QASs exploit in order to answer the user questions can be documents, data, and combinations of document and data [71].

Documents: the document-based knowledge sources have widely always received massive attention from by considering information retrieval techniques and recently deep neural network techniques [76]. The documents include plain texts, e.g., textual excerpts from Wikipedia, and require employing massive natural language processing and natural language understanding techniques to accomplish the goal of QASs [71].

Data: in the case of data, the underlying knowledge source exploits structured data. Since the amount of available structured data on the web consisting of RDF datasets keeps growing, RDF knowledge graphs have become a powerful asset for enhancing QASs. RDF knowledge graphs are huge collections of interconnected entities enriched with semantic annotations [77]. Due to the semantic relations of the data stored in knowledge graphs, applying complex natural language processing techniques is less than the document knowledge sources. Prominent examples of large-scale knowledge graphs include DBPedia [78], Yago [79], Freebase [80], Wikidata [81] Google's Knowledge Graph, Facebook's Graph Search, Microsoft's Satori, and Yahoo's Knowledge Graph [40,82].

Combinations: the combination of documents and data can be considered the hybrid knowledge representation. In this case, QASs target deriving answers using both a corpus and a knowledge graph [83]. Knowledge graphs are mostly incomplete, which leads to low recall values in QASs over KGs. On the other hand, the diversity of natural language makes document-based QASs difficult however a corpus may contain more answers than a

knowledge graph [84]. Thus, the combination of knowledge graphs and documents (or corpora) can improve the accuracy of QASs, e.g., GRAFT-Net [84] and PullNet [83].

### 5.4. Common Methodologies

The research progress of QASs as a reasonably long-existing research field of computer science can be categorized in three groups [7]:

1. Traditional techniques. Frequently asked question and answer (FAQs) and rule-based methods can be assumed as the major traditional and straightforward approaches. In FAQs, a set of question and answer pairs are collected and stored as the dataset. Then answers are generated by searching the given question from the stored dataset. When the required query is found, the relevant answer is given back to the user [85]. The rule-based methods primarily used in QASs over knowledge graphs rely on pre-defined rules or templates to parse questions and provide logical forms [7]. The definition of rules or templates leads to limited scalability and the need for researchers to become familiar with linguistic knowledge.

2. Information retrieval-based techniques. Machine learning, more specifically deep learning, plays a key role in many aspects of information retrieval systems [86]. A QAS over documents generally involves question and document representation steps, followed by a matching step to estimate the mutual relevance of the query and the document representations. A neural approach can affect one or more of these steps. In the case of QA over knowledge graphs, for a given natural language question, the named entities that reflect the main focus of the question are identified. Then the link between the extracted entities and the knowledge graph are specified. In the next step, a subset of the knowledge graph around the identified entities are extracted what its nodes are assumed as candidate answers. Based on the features extracted from the questions and candidate answers, the matching scores between the encoded answers and questions are calculated, and the final answer is selected. From the feature representation's perspective, the information retrieval-based techniques are divided into two groups including feature engineering and representation learning. In feature engineering, features are manually defined according to the questions' syntax information which basically fail to capture the semantic information of questions. In representation learning, questions and candidates are transformed to embeddings and then the embeddings are leveraged to compute the semantic matching and find the answer. The state-of-the-art QASs employ neural deep networks (e.g., memory networks, Convolutional Neural Network (CNN), Long Short-Term Memory (LSTM)) to generate better distributed embeddings for questions and candidate.

3. Semantic parsing-based techniques. Knowledge graphs based QASs can leverage semantic parsing-based techniques in order to answer user questions. These methods usually transform natural language questions into executable queries such as SPARQL queries or intermediate query forms such as query graphs based on neural semantic parsing with high scalability and capability. Recent QASs take advantage of graphs to represent questions, namely query graphs. Basically, the query graphs can be generated based on predefined natural language processing rules or neural networks. Then, the query graph which basically have topology commonalities with knowledge graphs are mapped to the knowledge graphs to find the answers. Additionally, trees or high-level programming languages can be used to represent questions through sequence-to-sequence models and attention mechanisms [87].

### 5.5. Categories

QASs can be classified according to a wide variety of criteria. Among the various possible categorizations, this section presents a description of the categorizations based on the application domain and forms of answers generated [70,72,88].

According to the application domain, QASs can be grouped into two groups: open-domain and closed-domain QASs. Open-domain QASs focus on answering domain-

independent questions within a huge knowledge source. In closed-domain QASs, the QASs answer questions under a restricted domain such as temporal, geo-spatial, medical, patent, and community [70,72,89].

The categorizations of QASs based on forms of answers include [14]:

1. Sentence/paragraph-answer based QAS: To answer some types of questions such as factoid or hypothetical ones, QASs return a single fact or a small piece of text as sentence/paragraph as the relevant answer.
2. Yes/no-answer based QAS: User questions are generally answered in the form of yes or no through verification and justification.
3. Multimedia-answer-based QAS: Answers are generated in different types of multimedia such as audio or video.
4. Opinionated-answer-based QAS: This system gives a star rating to an object as the answer.
5. Dialogue-answer-based QAS: These QASs are also known as dialogue systems that make efforts to answer users' questions in the form of a dialogue.

*5.6. Summary*

Many QASs have been emerging over documents and structured knowledge bases. According to the underlying knowledge sources, the process model of QASs can be classified into document-oriented model and data-oriented model. The main modules of the process model of QASs over documents include question analysis, passage retrieval, and answer extraction. Similarly, question analysis, question mapping, scoring and joint interface, query construction, and answer representation are assumed the basic components of the data-oriented process model. The text corpus and structured RDF data have been taken into consideration regarding data preparation and presentation in different QASs. The major techniques to answer natural languages questions consist of traditional techniques, information retrieval-based techniques, and semantic parsing-based techniques. Categorizing QASs based on domain involves open-domain and close-domain. The answer type criteria comprise sentence/paragraph-answer, yes/no-answer, multimedia-answer, opinionated-answer, and dialogue-answer-based QAS.

Although QASs have a long history and many approaches have been introduced, the accuracy of QA over unstructured and structured data still needs to be improved, and a significant amount of work is required to make them more beneficial and practical in the real world.

## 6. Dialogue System

Basing on an essential skill of human beings, interaction via natural language is the most straightforward way humans can access information on the web. If humans can communicate through natural language, they can conveniently access the desired information [90]. Dialogue systems, as a subset of QASs intend to communicate with human users relatively naturally through a dialogue. Dialogue can be considered a conversation between humans and machines. The form of communication can be text, speech, images, video clips, and or sign language [10,19,91]. Since these systems have to be interactive, incremental dialogue processing is required to generate continuous dialogue. In incremental processing, processing basically starts before the input is complete and that the first output is generated as soon as possible [92,93]. Apple Siri (https://www.apple.com/es/siri/, accessed on 1 July 2022), Google Assistant (https://assistant.google.com/, accessed on 1 July 2022) or Amazon Alexa (https://www.amazon.com, accessed on 1 July 2022) are some example of dialogue systems (more precisely chatbots) to support information search interactions. The dialogue systems can be applied to a wide variety of fields, including information-searching services via questions, virtual assistants to help users in daily tasks, e.g., scheduling appointments, and E-learning dialogue systems, e.g., train military personnel in questioning a witness [94]. For example, a goal oriented dialogue system for booking a restaurant would take several steps. First of all it would ask you what kind of food you

desire, where you are located, what price range is acceptable, and the date and time you like to book a table. Afterwards it suggests you some restaurants. Finally it asks you if you are satisfied or want to make some changes.

### 6.1. Current State of the Art and Related Works

Currently, the fields of dialogue systems range from small turn-based chat applications to large systems which can determine the emotions of the user and respond corresponding like Microsoft's XiaoIce [95]. There are also specific dialogue systems like the "Multimodal dialogue system with sign language capabilities", which has been proposed for deaf people [91].

The survey presented by Chen et al. [8] explains recent advances from deep learning to build dialogue systems and discusses possible research directions for task-oriented and non-task-oriented dialogue systems in detail. According to the survey, end-to-end-based and pipeline-based approaches are the main directions for task-oriented dialogue systems, while neural generative models, retrieval-based models or a mixture of neural generative and retrieval-based solutions are considered the major approaches to developing non-task-oriented dialogue systems.

Mallios et al. [96] present a high level process model for dialogue systems with six main components: automatic speech recognition, natural language understanding, dialogue management, knowledge base, response generator, and text-to-speech synthesis. Additionally, it proposes a classification scheme of dialogue systems, including answering systems, semi-dialogue systems, and full-dialogue systems.

Chaves et al. [97] show an increase in users' expectations for formal and specialized discourse in using conversational agents. The findings contradict the results presented by Liebrecht et al. [98], who argue that users may not assign different roles to conversational agents in a human-agent customer service setting.

### 6.2. Process Model

Different dialogue systems may have different process models but in general the various models can be summed up in three layers as shown in Figure 6 [19,95].

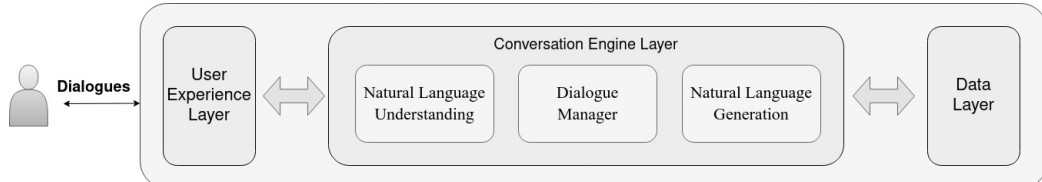

**Figure 6.** The process model of Dialogue systems.

1. User experience layer: this layer is the user's input and the dialogue system's output layers. As mentioned previously, the input and output can be in the form of voice, text, images, video clips, and or sign language. The user experience layer can be connected to many platforms such as Facebook Messenger, WhatsApp, WeChat, Sina Weibo, Tencent QQ, Telegram, and Skype.
2. Conversation engine layer: the second layer is connected to the other layers, including the user experience and data layers. It analyses the user input and preserves the data in a useful manner in the data layer. Depending on the task of the dialogue system, the data is used to give the desired answer. For speech-based dialogue systems, a Text To Speech module is added at the end. The conversation engine layer consists of three main modules as shown in Figure 6 [19].

    (a) Natural Language Understanding Module: it takes a user query in a natural language manner and translates it into a semantic representation. If the input is not a text, it is converted to a string of words, then stored in the Data Layer.
    (b) Dialogue Manager Module: the semantic representation of the user's text is taken, and a schematic representation of response is generated.

Indeed, the module keeps track of the dialogue state and a dialogue policy learner, which decides what to answer to the user.

(c) Natural Language Generation Module: the semantic representation of the answer is translated to a natural language answer. Thus, it makes decisions regarding the information to be included, how information should be structured, choice of words, and syntactic structure for the message.

3.  Data layer: all data, including collected conversational data, non-conversational data, and knowledge resources, is stored in the data layer as a set of data sources.

### 6.3. Data Preparation and Representation

Dialogue systems intend to interact with human beings and rely on a huge amount of different types of data, including unstructured, structured, and semi-structured data [8]. The key ability of these systems is to understand unstructured data as input and interpret it correctly and then provide the output across the underlying knowledge base, which are described as the following:

1.  Unstructured data: this kind of data is not organized in a predefined manner. Text documents are the most well-understood instance of unstructured data which do not have any predefined data model.
2.  Structured data: due to addressable elements in structured data, it is easy to be analyzed. It can be effectively organized in a formatted repository such as databases. The relational databases are the most well-known model for holding tabular data as an example of structured data.
3.  Semi-structured data: it lies between unstructured and structured data. However, the semi-structured data does not reside in a relational database, but it has some organizational features that make it easier to analyze. The XML data resources can be considered semi-structured data.

### 6.4. Common Methodologies

According to Chen et al. [8], different methodologies to develop dialogue systems can be summarized into the following varieties:

1.  Pipeline methods: the main idea behind the pipeline methods is to define a pipeline structure including natural language understanding, dialogue state tracker, dialogue parser learning, and natural language generation. Natural language understanding is responsible for detecting the intent of users through classifying the intent into predefined intents or employing some techniques such as deep learning. The dialogue state tracker handles the input of each turn along with the dialogue history and returns the current dialogue state. Then the dialogue parser learning learns the following action based on the output of the dialogue state tracker. Finally, the response is returned based on the action [8].
2.  End-to-end methods: the task-oriented dialogue systems can apply an encoder-decoder model to train the whole system. The model only adopts a single module and interacts with structured external databases. In end-to-end methods, dialogue system learning can be viewed as the problem of learning a mapping from dialogue histories to system responses [8].
3.  Retrieval-based methods: the major focus of retrieval-based methods is message- response matching by employing matching algorithms to bridge semantic gaps between messages and responses. So based on the matching, a response from candidate responses is selected [99]. The early retrieval-based dialogue systems apply single-turn response matching. In these systems, only the message is used to select a proper response in each single-turn conversation, while multi-turn response matching has been mainly used in recent years, current messages and previous utterances are taken to select the response. Thus, the selected response is natural and relevant to the whole context [8,100].

4. Neural generative methods: to develop non-task-oriented dialogue systems such as chatbots, neural generative methods can be basically adopted. The sequence-to-sequence models as the foundation of generative methods lead to keeping conversations active and engaging [8,101].

5. Hybrid methods: recently, the neural generative and retrieval-based models have been combined to increase performance. Since the retrieval-based systems generally return accurate but straight responses, while generation-based systems often return fluent but meaningless responses, the combined approaches can have significant effects on performance [8,102,103].

Neural models such as CNN, Recurrent Neural Network (RNN), Hierarchical Recurrent Encoder-Decoder (HRED), memory networks, attention networks, transformer, deep reinforcement learning models, and knowledge graph augmented neural networks are mainly used in state-of-the-art dialogue systems [104], but this paper focuses on providing a generic picture from the methodologies and methods' prospective.

Recently, dialogue systems apply CNNs as a hierarchical feature extractor after encoding the dialogues [105,106]. RNN models and their variants (e.g., gated recurrent unit (GRU), LSTM) as powerful learning models are utilized in dialogue-related tasks since the dialogues are not independent of each other and are not of fixed length [107]. HRED as context-aware sequence-to-sequence models are able to capture hierarchical dialogue features in dialogue systems [108]. To keep dialogues, memory plays an key role in dialogue systems. That is the reason why memory networks are fulfilled in these systems especially task-oriented systems [104]. Attention networks and transformers are employed in the state-of-the-art dialogue systems due to attention is able to catch the importance of different parts in the dialogues and transformer not only is a sequence-to-sequence model but also is a model to represent the dialogues [109]. Additionally, due to the agent-environment nature of dialogue systems, deep reinforcement learning models are applied to make improvements in these systems [110]. Knowledge graph augmented neural networks are basically used in the dialogue systems that rely on structured or semi-structured data such as knowledge graphs [111].

### 6.5. Categories

Conversational or dialogue systems are categorized into task-oriented and non-task-oriented groups [10]:

1. Task-oriented dialogue system: Furthermore, known as goal-oriented dialogue system, it targets to assist users in completing tasks in one or multiple domains by the end of the dialog. Some goal-oriented dialogue systems include restaurant reservation, flight ticket booking, and course selection advising, finding products. The major methods to develop task-oriented dialogue systems include the pipeline or end-to-end methods.

2. Non-task-oriented dialogue systems: It also called chatbot can be defined as software to provide extended conversations and mimic the unstructured conversations or 'chats' characteristic of human-human interaction [10]. Chatbots aim to maximize user engagement and interact with a human to provide reasonable responses and entertainment. Typically they focus on conversing on open domains. However, the none-task-oriented systems have been basically designed for entertainment. They also intend to gain practical targets over time like task-oriented chatbots [8,10]. The widely applied method to none-task-oriented dialogue systems are the retrieval-based, neural generative-based, and hybrid methods.

### 6.6. Summary

Dialogue Systems as a subset of QASs allow the user to converse with a machine using natural language. The major layers of a dialogue system include the user experience layer, conversation engine layer, and data layer. The conversation engine layer consists of the natural language understanding module, the dialogue manager module, and the natural language generation module. The data layer relies on a massive amount of various sorts of

data, including unstructured, structured, and semi-structured data. The research progress of dialogue systems can be summarized into pipeline methods, end-to-end methods, retrieval-based methods, neural generative methods, and hybrid methods. The task-oriented and non-task-oriented dialogue systems such as chatbots are the most significant types of dialogue systems.

## 7. Chatbot

Chatbots are the most straightforward kind of dialogue systems designed for imitating human-human interaction [10]. Areas that significantly benefit from chatbots are entertainment, health care, marketing, supporting system, and customer service. Especially in the customer service domain, chatbots reduce costs by handling multiple users simultaneously with a 24/7 availability.

Imagine a chatbot embedded on a social media platform helping users book a table in your restaurant. Based on the information provided by your profile, the chatbot needs only a minimal amount of additional information to fulfill the table reservation. For example, the user's name can be taken from the profile, and only the desired date and time need to be provided. If successful, the chatbot immediately returns a confirmation. Besides, the chatbot may send a notification when the user has to leave to reach the restaurant in time (based on the current location, if the user uses an app and allowed location features).

The era of chatbots started a while ago with ELIZA [112] in 1966, then PARRY by Kenneth Mark Colby in 1972 [113], the first chatbot with personality, and the well-known intelligent digital assistants Amazon Alexa, Apple Siri, Google Assistant, or Microsoft Cortana, we use nowadays [114]. The main idea behind chatbots is the automation of services via conversational interfaces. Chatbots may be not only a tool but also a friend. However, they still struggle to understand the meaning and undertones of conversations.

Nowadays, most of the existing chatbots are acting on a closed domain [115]. Closed domain implies that they are designed for a particular task, e.g., booking a hotel room or reserving a table at a restaurant. Examples of open domain chatbots are Amazon Alexa, or Apple Siri, able to answer all kinds of questions [115].

Since we are in the middle of a pandemic, chatbots are also usable for sharing up-to-date information quickly and concisely. In such a scenario, short and precise answers from a trustworthy resource are essential to answer all kinds of questions [116].

### 7.1. Current State of the Art and Related Works

Chatbots have a pretty long history starting with the first chatbot ELIZA in 1966. Since then, many new approaches for designing and implementing chatbots have evolved. An overview of the history of chatbots is given in the work presented by Weizenbaum [112].

In addition to the history of chatbots, the surveys presented by Jurafsky et al. [10], Nimavat et al. [115], Adamopoulo et al. [117] and Deshpande et al. [118] classify chatbots based on different characteristics, be it the knowledge domain, their goal, or the communication medium. Besides, essential concepts and methodologies for processing natural language input and natural language generation are introduced. Every chatbot follows a general process model of user input, input processing, dialogue management, chat engine, and response generation. Suta et al. [119] cover the same topics focusing more on machine learning approaches used by chatbots.

From the technology anxiety perspective, the results of studies by Kim et al. [120], Lee et al. [121] and Yang et al. [122] show that technology anxiety is a reducing factor for relationships between the chatbot quality dimensions (i.e., understandability, reliability, assurance, and interactivity) and users' technology adoption. However, studies by Chin et al. [123] and Li et al. [124] argue that the higher levels of technology anxiety, the stronger the relationships between the chatbot quality dimensions and users' technology adoption.

For the design and implementation of chatbots, the user experience is an essential part. In the study conducted by Jain et al. [125], first-time chatbot users are interviewed about

their experience using different kinds of chatbots. As an outcome, users prefer human-like natural language chatbots or chatbots with an engaging experience. Those findings are usable for the design of new chatbots.

### 7.2. Process Model

In general, the process model of chatbots consists of four major components: Natural Language Understanding (NLU), Dialog Management (DM), Chat engine, and Response Generation [10,115,117,118]. Figure 7 presents an overview of the process model and how the components are connected.

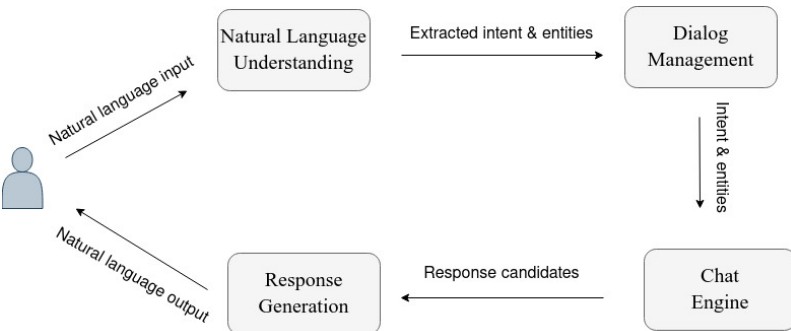

**Figure 7.** Chatbot process model.

Everything starts with a user input that needs to be processed by the system. The user input reaches the NLU component to be analyzed. In this component, the input is parsed, and the user intention and associated information are extracted. Besides, an NLU might analyze a user's sentiment, positive, negative, or neutral [119]. The output of this component is the identified intention (what the user wants to achieve) and associated information. For example, the intention of the question "What is the weather going to be in Innsbruck tomorrow?" is to get weather information. Associated information is the location ("Innsbruck") and the date ("tomorrow"). That information is passed to the Dialogue Management component to be processed further. The Dialogue Management Component keeps track of the current conversations, and their contexts [115,117]. This implies the current intent and identified entities. Besides, this component ensures that all required information is available, follow-up questions are sent to the user if any information is missing. When all information is given, they are forwarded to the chatbot engine.

Each chatbot's core is the chatbot engine, responsible for executing the user's intended action. The underlying data, be it a database or an external API, is accessed to retrieve the information sent to the user. Based on the found information, single or multiple possible responses are sent to the Response Generation.

The Response Generation component uses Natural Language Generation (NLG) approaches to construct a personalized response using natural language based on the responses. Personalized means a proper writing style and emotions [119]. Additionally, the response should be grammatically correct, and the user should not recognize that the opponent is a chatbot rather than a human [126]. Therefore, the response should also include some emotions and imitate human-like behavior [127].

### 7.3. Data Preparation and Representation

The main goal of chatbots is to either help a user fulfill a particular task (closed domain) or imitate human-human conversations (open domain). Both ways need data that can be given to the user. Data used to power chatbots can be either unstructured, semi-structured, or structured [119]. Some chatbots store and host the data on their side, whereas others retrieve the relevant information in real-time from APIs. Retrieving APIs in real-time requires a connection from the chatbot engine towards the service to access. The connection to those web services can be hardcoded for simpler chatbots that require access to a small

number of APIs only. For chatbots handling large amounts of web services or when web services change very often, a description language on how to map the incoming parameters from the NLU component to the request's parameters to the web service is required. Web API Annotations with Schema.org Actions (WASA (http://wasa.cc/, accessed on 1 July 2022)) is such a description language. WASA is based on schema.org actions allowing to describe web services as a set of actions to be taken (e.g., search).

Storing chatbot data requires a database, such as a document store, a relational database, or knowledge graphs repository. A knowledge graph is an optimal solution for more complex chatbots accessing a vast amount of knowledge [59,128]. Due to the flexibility of knowledge graphs, integrating heterogeneous sources is not an issue [59]. A vocabulary for defining the data stored in a knowledge graph is often schema.org [59]. Schema.org is the de facto standard for annotations on the web founded by the large search engine providers, such as Google, Bing, Yahoo!, Yandex.

### 7.4. Common Methodologies

Standard methodologies for chatbots mainly concern intent detection and extraction of entities. Besides, answer generation is essential for chatbots. In the following, methodologies and algorithms based on the classification presented by Hussein et al. [9] are considered:

1.  Parsing—This category extracts meaningful information from the textual input of the user. The grammatical structure of a sentence is used to extract keywords that are then matched against the stored data to find the appropriate response. Semantic parsing is a more advanced technique for converting the input sentence into a representation that machines can process. For example, Dialogflow recognizes relevant information based on a predefined set of training phrases containing placeholders for parts of the sentences containing relevant information. Those placeholders identify a specific type of information (e.g., location information or name information) that can be used to generate queries to the underlying database by using templates. This approach allows an explicit definition of what happens with the relevant information. However, it requires a lot of manual work to set up the chatbot properly [9].

2.  Pattern Matching—This approach is most commonly used. It defines a set of hand-crafted pattern-template pairs. Whenever a pattern matches the input, the template is used to return a response to the user. This approach is mainly used in QA chatbots and is very flexible for creating conversations. However, all possible patterns are built manually, which is not scaling. Due to the scaling issues, the responses are predictable and are not manifold, more repetitive [9].

3.  Ontologies—Ontologies are used to represent domain knowledge and make it explorable by the chatbot itself. An advantage of this approach is that the chatbot can use reasoning to detect relationships between concept nodes used in the conversation [9].

4.  Markov Chain Model—The Markov chain model is a probabilistic model modeling probabilities of state transitions over time. There is a fixed probability for the following states based on a given state. A chatbot using this model produces outputs based on those state transitions. The chatbot constructs probabilistically more suitable responses. This model is mainly used for chatbots that entertain users by imitating simple human conversations. Markov chain models however do not work well on complex conversations [9].

5.  Neural Networks Models—These models allow more intelligent chatbots. The research trend for using artificial neural networks for chatbots is a generative-based approach where chatbots dynamically generate the response to the given user input. Neural network models are learning algorithms used in machine learning and can be supervised and unsupervised. Subclasses of artificial neural networks models used for chatbots are RNNs, sequence to sequence neural model, or LSTMs. A major problem with these artificial neural networks is that they are not accurate yet.

Therefore, prominent virtual assistants like Alexa, Siri, and Cortana rely on a semi-rule-based approach [115].

*7.5. Categories*

Chatbots can be classified based on different categories [10,115,117,118]. However, a chatbot does not entirely fall into a specific category but is part of multiple categories depending on different characteristics.

Starting with the build method, open-source platforms and closed platforms are distinguished [117]. Open-source platforms allow a broader community to develop the platform further and check how the platform is working. A closed platform acts as a black box, and further developments depend on the company maintaining the platform. As a communication medium [119], chatbots can either be integrated on messaging platforms (e.g., Facebook Messenger, Slack, WhatsApp) or used on smart home devices (e.g., Amazon Alexa, Apple Siri, Google Assistant) as so-called skills.

We follow with moving away from technical aspects to the concept behind the chatbot themselves and its goals. For goals [117], chatbots can be classified as informative, chat-based, or task-based. Informative implies that users can ask for information about specific topics or get general information (e.g., health information during the corona pandemic [116]). Chatbots with a chat-based or conversational goal act like another human being imitating real conversation with users. These can be seen as a friend of the user. Task-based chatbots are designed to perform particular tasks, e.g., a chatbot on a restaurant website that mainly handles table reservations. Usually, such chatbots also provide some information about the restaurant, but their main task is to allow users to reserve tables.

Some chatbots do not act autonomously but include a human in certain parts. This category is referred to as "Human-aid" [117]. A chatbot falls into this category if at least one part of the chat flow involves a human.

The core functionality of a chatbot is the processing of inputs and the generation of responses [117]. In this category, several types are distinguishable. The first chatbots (ELIZA and PARRY) were based on pattern matching approaches. A given input was matched according to the inputs stored in the internal database, and the predefined response for the matching input was returned. Disadvantages of this approach are that there will always be the same answer for the questions, no human touch, and previous conversations are not stored. Rule-based approaches are pretty similar to pattern matching. Based on a fixed set of rules, responses are generated. User input is mapped to the associated responses. Retrieval-based chatbots do not store the data themselves but query and analyze external sources using APIs. For example, a weather chatbot consults the API of predefined weather services. Generative or corpus-based systems are trained on a vast dataset of human-human conversations. Answers are generated based on answers from previous conversations. This approach is more human-like, uses machine learning algorithms and deep learning techniques but is challenging to build and maintain. Especially for the setup of such bots, a vast amount of training data is needed. Besides the previously mentioned types, there are hybrid approaches that combine rules and machine learning approaches. In such cases, the management of the conversation flow is defined by rules, and the responses are generated using natural language processing approaches.

Regarding the knowledge domain [115,117], two categories are used to classify chatbots. Closed domain chatbots focus on particular topics or specific tasks. Those chatbots maybe have some small talk content but mainly focus on the topic for which they are designed. Open-domain chatbots try to cover as many topics as possible.

Finally, chatbots can be distinguished based on the service they provide [115]. Suppose a chatbot is designed for a specific task, be it reserving a table in a restaurant or more as a communication medium to automate services. In that case, they are classified as interpersonal. Those chatbots do not act as friends of a user but more as a tool to achieve a specific goal. Intrapersonal chatbots are designed to be a friend of a user. They should understand the user like humans do. In the third category, there are inter-agent chatbots.

Here, a protocol for the communication between chatbots is needed. An example of this type is the Alexa-Cortana integration.

*7.6. Summary*

Chatbots are a subclass of dialogue systems and the most superficial subtype of those. The process model can be split into four major components: the NLU component that identifies the user intention and extracts the entities, the Dialogue Management component taking care of the conversation flow and follow-up questions when information is missing, the chatbot engine itself accessing the underlying database or external API to retrieve the responses that are relevant for the user, the NLG component creating a response for the user in a way a human would communicate. The presented methodologies mainly target NLU and NLG since those are the most crucial parts of a chatbot.

However, there are still many issues that need to be tackled by further research to make chatbots more human-like. This includes context awareness, diversity of responses, and chatbots should get a stronger personality.

**8. Future of Search on the Web**

As the primary tool for users to access information, the importance of search on the web can not be overestimated. To sum up, the main limitations or challenges of each type of the described search services are summarized as following:

1. Keyword search engines fail to (1) disambiguate words and return irrelevant results (false positives), and (2) turn up related materials that do not specifically use the search keywords (false negatives).
2. The stability of the semantic web languages and continuous development of ontologies are key challenges in semantic web search engines. Furthermore, the performance of semantic web search engines relies on knowledge sources, which might contain errors and missing values.
3. Despite a lot of progress in QASs, lexical gap and ambiguity are still main challenges in these systems either documents-based, data-based, or mixture of document and data based knowledge sources.
4. In addition to lexical gap and ambiguity, dialogue systems need to tackle with incremental processing due to these systems are naturally interactive.
5. Additionally to the above challenges, remembering the context of a conversation (or even understanding in some cases) needs to be improved in chatbots.

Although keyword search is still important, semantic search is on the rise since a search query aims to not only find query terms but to determine the intent and contextual meaning of the terms. Interacting with computers through QASs, dialogue systems and chatbots is expected to take over searching on the web because of conversational artificial intelligence and great potential and commercial values [129]; dialogue systems and chatbots will reduce the number of questions needed to reach answers.

Recent advances in machine learning and natural language processing will aid search on the web to improve different aspects of the interaction-based search services (i.e., QASs, dialogue systems and chatbots) and understand different ways users naturally interact and interpret information. In consequence, search on the web will manage multiple varieties of interaction between users and search services and users will be able to express their queries in different ways, including text, image, audio, video, or any combination of them. For example, a single search query can be expressed as a combination of text and image [130].

Furthermore, multilingual search will be improved in order to return result for a search query which is expressed in a language different from languages of underlying resources. Search on the web will become more personal and move from returning content from resources towards seeking resolution for users' problems [131]. They may be capable of actively predicting user questions before searching them, providing the users with results they probably need before they truly need and search them [132]. So contextual understand-

ing and personalization for the users would definitely be from the main challenges [132]. Recall the example of food order, users currently take advantage of search services to search and order their foods while it can be imagined that foods will search users in the future, employing the web and proactively, basing on the nutritional condition, context and preferences of the user. With levels of personalization and contextualization, the search will become more conversational and easier in the future.

With regarding such visions of the future, today's search services are still a simple tool, and definitely, there are still many opportunities to make much progress and improve [133]. There are also many challenges to handle, for example, ensuring that the ongoing developments have a positive impact on society as a whole, mitigating negative impacts and ensuring positive impacts for everyone from the big data technology adoption [134]. This is applicable also to the search on the web as an important activity most of us execute on a daily basis.

## 9. Discussion

Throughout this research study, we provide a generic analysis of different services to find information on the web as one of the primary information resources. We start with categorizing services that have been introduced or developed to manage the growing amount of information and data accessible on the web. Here, the search services are systematically summarized and categorized into five groups, including keyword search engines, semantic search engines, QASs, dialogue systems and chatbots. Table 1 summarizes the evolution process. Then, we give an overview of the state-of-the-art around search services and highlight the relevant aspects of the services.

To compare the introduced search services, Tables 2–7 provide their details on the timeline, approaches and tools, typical usage, pros, cons and open challenges in each search service, respectively.

**Table 1.** Evolution among different search services.

| From | To | Description |
|---|---|---|
| Keyword search engine | Semantic Search Engine | Keyword search engines look for literal matches of the query words, while semantic search engines not only return results based on keywords but also consider the contextual meaning and the user's intent when fetching the relevant results. |
| Semantic search engine | Question answering | Semantic search engines fetch relevant results based on the user's query, while QASs allow the user to submit a question and derive the pertinent and exact answers. |
| Question answering | Dialogue system | The objective of a QAS is to focus on producing the exact answer for a single-user question, while unintentionally ignoring the reasons that motivated that user to pose this question. So, dialogue systems as conversational agents are developed to extend conversations between the user and agent. |
| Dialogue system | Chatbot | Chatbots are a sub-type of dialogue systems that perform chit-chat with the user or serve as an assistant via conversations. |

**Table 2.** Timeline.

| Keyword Search | Semantic Search | Question Answering | Dialogue System | Chatbots |
|---|---|---|---|---|
| The first keyword search engine called "Archie" was launched in 1990 [22]. For almost two decades, it was the most common approach that supported browsing services on the web [22]. Keyword search engines were gradually replaced by semantic search engines in the early 10's [135]. | Semantic search has emerged with the semantic web in the early 00's [43]. However, the major breakthrough of the semantic search engines as a service was in 2013, when Google presented the "Hummingbird" update [135]. As of today, browsing services supported by semantic search engines play a crucial role in the big picture of the web as a service. | The first QAS was built in 1961 to answer simple questions relating to American League baseball games [136]. With emerging knowledge graphs in 2012, QASs across knowledge graphs have been introduced. Since the early stage, QASs have advanced significantly, particularly about a decade ago, due to progress in natural language understanding and deep neural networks | The concept of conversing in language shifted from fiction to scientific pursuit when Alan Turing forecast machines that could converse like humans in 1949 [137]. Dialogue systems took significant steps in 1966 and 2011 with the emerging ELIZA and Siri, respectively [94]. Work on dialogue systems has been of broad significance in the recent decade. | In 1966, ELIZA, the first chatbot, appeared intending to imitate a psychologist [138]. PARRY in 1972 was another famous chatbot 16 years later [113]. The first chatbots were relatively simple, and nowadays, they are much more intelligent. Examples of intelligent digital assistants (as they are called nowadays) are Amazon Alexa, Apple Siri, Google Assistant, and Microsoft Cortana [114]. |

**Table 3.** Approaches/Tools.

| Keyword Search | Semantic Search | Question Answering | Dialogue System | Chatbots |
|---|---|---|---|---|
| • Web content mining.<br>• Web usage mining.<br>• Web structure mining. | • Transform unstructured and semi-structured data into structured knowledge.<br>• Schema alignment, entity matching, and entity fusion<br>• Entity, relationship, and document ranking.<br>• Content negotiation to interact with structured knowledge. | • Traditional techniques.<br>• Information retrieval-based techniques.<br>• Semantic parsing-based techniques. | • Pipeline methods.<br>• End-to-end methods.<br>• Retrieval-based methods.<br>• Neural generative methods.<br>• Hybrid methods. | • Parsing.<br>• Pattern matching.<br>• Ontologies.<br>• Markov chain models.<br>• Neural network models. |

**Table 4.** Typical Usage.

| Keyword Search | Semantic Search | Question Answering | Dialogue System | Chatbots |
|---|---|---|---|---|
| For users typing (e.g., "Restaurants" and "Innsbruck"), Keyword Search Engines return web pages containing these keywords. The users have to check the returned pages one by one to find relevant information for them. | For the case of "Pizza, a semantic search engine might return a Pizza description, which details its definition, name, nutritional facts, and local where one can order it. | For users' questions (e.g., "Which restaurants in Innsbruck serve Austrian cuisine on Sundays for between €30 and €60?"), QASs provide precise answers. | Users' intents are identified through a dialogue (e.g., to book a restaurant's table, a dialogue is shaped to understand the user's information needs such as kind of food, location, price range, etc.). | For the case of booking a table in a restaurant, users' personal information is used to return a restaurant that matches the user's preferences and there is no need for the users to leave the chatbot and call the restaurant. |

**Table 5.** Pros.

| Keyword Search | Semantic Search | Question Answering | Dialogue System | Chatbots |
|---|---|---|---|---|
| • Pave the way for the next generation of search engines. | • Improve search accuracy by consuming machine-readable content from websites and structured knowledge from knowledge bases. | • Submit questions in a natural language form.<br>• Supply direct answers to questions. | • Context-awareness due to responding based on the current message and the conversation history.<br>• Response coherence due to maintaining logic and consistency in a dialogue.<br>• Interactive training due to improving themselves via interactions with users. | • Communicate in natural language.<br>• Include personal user information for better search results<br>• Help users in achieving a certain goal (e.g., booking a table in a restaurant) in goal-oriented chatbots.<br>• Interact with external services (e.g., table reservation, hotel room booking). |

**Table 6.** Cons.

| Keyword Search | Semantic Search | Question Answering | Dialogue System | Chatbots |
|---|---|---|---|---|
| • False positive due to polysemy words.<br>• False negative due to synonym words.<br>• Be literal and ignore the context of words as well as the intentions of users.<br>• Ineffective information retrieval process in large-scaled resources. | • Complex implementation.<br>• Limited powerfulness due to the knowledge acquisition process.<br>• Rely on the quality of knowledge sources. | • Lack of conversation to verify users' intents in complex questions.<br>• Lack of dialogues to fulfill all aspects of information needs once users' requirements can not be expressed using a single question and requires the user to ask multiple questions. | • Privacy threats due to serving many users, having the ability to learn through interactions and storing sensitive information. | • Lots of manual work in formulating training phrases.<br>• Low accuracy for neural network models with no predefined training phrases.<br>• Low response diversity. |

**Table 7.** Open Challenges.

| Keyword Search | Semantic Search | Question Answering | Dialogue System | Chatbots |
|---|---|---|---|---|
| • No significant open challenge. | • Build more scalable semantic search engines according to the growth of the semantic web.<br>• Curate and update knowledge bases to offer high-quality knowledge due to the heavy dependence of semantic search engines on the knowledge bases to deliver information. | • Understand complex semantics in input questions.<br>• Reduce huge search space in knowledge bases.<br>• Incomplete knowledge bases. | • Incremental processing and swift warm-up.<br>• Perceive and express emotions.<br>• Privacy protection. | • Context awareness.<br>• Diversity of responses (natural language generation).<br>• Chatbots with a stronger personality. |

## 10. Conclusions

This paper overviews different types of search possibilities on the web that help users meet information needs. From the beginning of the web to the present day, keyword-based search engines, semantic-based search engines, QA systems, dialogue systems, and chatbots have been proposed to explore the web. Keyword-based search engines have been the first generation of search engines, and then semantic search engines have emerged to solve the limitations of the keyword-based search engine. QASs, dialogue systems (as a subdivision of QASs) and chatbots (as a subdivision of dialogue systems) are targeting to assist users find pertinent information through a natural language interaction. We analyze the current state and related works, process model, data preparation and representation, methodologies, and categories for these types of search services in our research study. We also show that the development of the search on the web, however, is still an ongoing process. The future systems will focus on multiple varieties of interaction and multilingual search. Those systems are expected to deliver answers to users even before getting the questions asked by the users however such and similar developments comprise societal, business and ethical issues that need to be considered when designing the search on the web of the future.

To the best of our knowledge, this survey is the first, most comprehensive and up-to-date one currently covering all the web search possibilities throughout the years. This work helps academics and developers who need to grasp all the search services quickly and understand their developments and evolution and also provides a good starting point for them to probe deeper into each of these services.

We also acknowledge that our study can not analyze all aspects of search services with technical discussions. A future research direction is to evaluate and compare the efficiency and accuracy of the retrieval results of the services [5]. Another possible direction for future work is to focus on a relatively small time span (e.g., the latest ten years) and deepen the discussions and analysis of the literature.

**Author Contributions:** Conceptualization: S.A.; Methodology, formal analysis, investigation: S.A., K.A., E.H., G.B., M.S. and A.F.; Writing—original draft: S.A., K.A., E.H., G.B. and M.S.; Writing—review and editing: S.A., G.B. and A.F.; Supervision, funding acquisition: A.F. All authors have read and agreed to the published version of the manuscript.

**Funding:** This work has been partially funded by the project WordLiftNG within the Eureka, Eurostars Programme (grant agreement number 877857 with the Austrian Research Promotion Agency (FFG)) and the project KI-Net within the Interreg Österreich-Bayern 2014–2020 programme (grant agreement number AB 292).

**Institutional Review Board Statement:** Not applicable.

**Informed Consent Statement:** Not applicable.

**Data Availability Statement:** Not applicable.

**Conflicts of Interest:** The authors declare no conflict of interest.

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
