# Peer review of "Interactive Search on the Web: The Story So Far"

_information, doi:10.3390/info13070324_

Round 1
Reviewer 1 Report
The present paper is a literature review survey on the field of search engines focusing on the different types of search services. It is comprehensive and can provide value to its readers.
Some suggestion are as follows:
It might be a good idea, when directly mentioning a study or research in a sentence, to use the researcher’s name alongside the reference number, in order to improve readability. For example line 131 should be “The history and rise of keyword search engines are studied by Seymour et al[15]” and line 136 should be “In Rahman’s survey[16] the major challenges, issues, and downsides of keyword search engines have been discussed in detail.” It is inherently counter-intuitive to read a sentence that the object or subject is just a reference number. Such mention is not necessary when the research isn’t directly mentioned as a sentence’s subject or object.
In the beginning of Section 3 detailing the keyword search service maybe a mention of metasearch engines and federated search would be in order. They are interesting variations especially in the data gathering field.
Despite the fact that the survey is comprehensive there are no mentions of contradictory findings from the various papers presented. It would create a more interesting read, if cases where researchers find contradictory conclusions or take contradictory approaches are discussed.
In the future section, a discussion concerning the popularity of the various presented services might be useful. Which services are on the rise and which are declining and why?
And some minor proofreading points:
Line 44 “the system converts”
Lines 242-243 are confusing. Please rephrase.
Line 451 “the technology that empowers semantic search engines is already in place”
Line 579 The usage of the word “Since” in the beginning of that sentence is confusing. Please rephrase.
Line 623 “a predefined natural language processing rules” Usage of “a” with plural is a mistake?
The level of English used in this paper is good, but some proofreading and rephrasing for clarification purposes might be needed.
Reviewer 2 Report
The paper proposes a categorization of the search engine services from the early days of the web until today. The theme of the study does seem to be compatible with the scope of the Digital journal.
The structure of the paper is quite logical. It is based on the different categories of the systems that were studied. Nevertheless I found a bit strange the summary sections that were incluses in each category. This of cource can be justified by the increased size of the manuscript.
Overall the paper offeres a nice overview of the various services on the WWW, although I can not say that the title of the paper clearly describes the content of the study. For example I would not expect chatbot to be included in the study although when reading the study I understand the relevance that all those included service have. I would recommend the authors to modify the paper title to something more representative of the content of the study. For example Interactive search on the web….
One issue that I have an objection to is that the authors have submitted the manuscript as a regular article, whereas I believe that this particular study is best described as an overview paper. One may argue that this paper is a systematic review but in this case, the authors should provide quantitative data on the number of scientific papers which were collected for each category of the study. The fact that this paper can be categorized as an overview paper is supported by the fact that the organization of the manuscript is similar to a book, which usually gives an overview of the state of the art in a certain scientific area.
Some other issues that should be addressed are:
Lines 21-22 – this is a 2019 prediction and not a fact.
Lines 27-32 – this section needs to be supported by references.
Lines 82-90 – Please specify the time frame that this survey covered and when it took place.
Table 2 includes a lot of dissimilar information, it spans 4 pages and is very difficult to read and comprehend. I suggest that the author can distribute the content to more tables in order to increase its readability.
In the case of chatbots, the authors focus only on a specific chatbot category that employs NLP. There are other types of chatbots something that is not visible in the analysis.
Overall I believe that this manuscript offers a unique, and systematic overview of an important group of web services and after the revision, it would be a useful contribution to the Information Journal.
Reviewer 3 Report
Search on the Web: The story so far
The topic is interesting and diagrams and tables are illustrative. Although the article has more sections than are usual in the scientific literature, the organisation of the manuscript is clear. Here are some recommendations to improve the structure and content of the manuscript.
Keywords: Three to ten pertinent keywords need to be added after the abstract. We recommend that the keywords are specific to the article, yet reasonably common within the subject discipline.
To increase the visibility of the article through bibliographic search engines, you could add other keywords like some of the methodologies used by search engines or the names of the current major search engines.
There is an important omission that needs to be addressed. References are missing from one of the world's most trafficked web search engines: Baidu.com. Please see “Measuring destination image through travel reviews in search engines” (MDPI, 2017).
“For instance, various standards, such as” RDF ==> Resource Description Framework (RDF)
“The first keyword search engine was called ‘Archie’” (Table 2)
The reference to corroborate this paragraph is missing. Please see Seymour et al. (2011). Also, on “In 1966, ELIZA, the first chatbot, appeared intending to imitate a psychologist.”
The format of Tables 1 and 2 is not optimised. Please see Table 2 of the “information-template.dot” document.
Conclusion
Concluding remarks section should have a summary of the main study outcomes and the way to obtain them, and three subsections:
Theoretical implications: Does the study contribute to the body of knowledge of web search engines from a theoretical perspective?
Practical implications: Can the study be useful for academics, developers or users of web search engines?
Limitations and future work: A simple study cannot cover all aspects of a subject as complex and extensive as web search engines. What could be the future lines of research in this area? For example, regarding the methodologies seen in this study, check the efficiency and accuracy of the main search engines. In different search engines, does the same request retrieve similar data? (MDPI, 2017).
The Author Contributions, Funding, Data Availability Statement, and Conflicts of Interest sections are missing.
Reference list
Please check the citations and references: Multidisciplinary Digital Publishing Institute (MDPI) style. I use Mendeley.com or Zotero.org with the MDPI template. For example: Abbreviated Journal Name.
The following reference is incomplete:
FROM:
Deshpande, A.; Shahane, A.; Gadre, D.; Deshpande, M.; Joshi, P. A SURVEY OF VARIOUS CHATBOT IMPLEMENTATION TECHNIQUES. 2017.
TO:
Deshpande, A.; Shahane, A.; Gadre, D.; Deshpande, M.; Joshi, P. A survey of various chatbot implementation techniques. International Journal of Computer Engineering and Applications 2017, 11
Round 2
Reviewer 2 Report
None
Reviewer 3 Report
Interactive search on the Web: The story so far
The manuscript has improved significantly and therefore I recommend its publication in MDPI Information.
Good luck!